# Reduced Cholinergic Activity in the Hippocampus of Hippocampal Cholinergic Neurostimulating Peptide Precursor Protein Knockout Mice

**DOI:** 10.3390/ijms20215367

**Published:** 2019-10-28

**Authors:** Yuta Madokoro, Yuta Yoshino, Daisuke Kato, Toyohiro Sato, Masayuki Mizuno, Tetsuko Kanamori, Masamitsu Shimazawa, Hideki Hida, Hideaki Hara, Mari Yoshida, Cesario V. Borlongan, Kosei Ojika, Noriyuki Matsukawa

**Affiliations:** 1Department of Neurology, Nagoya City University, 1 Kawasumi, Mizuho-cho, Mizuho-ku, Nagoya, Aichi 467-8601, Japan; yutamadokoro@yahoo.co.jp (Y.M.); daisuke@med.nagoya-cu.ac.jp (D.K.); toyohiro_st@yahoo.co.jp (T.S.); m_mizuno@med.nagoya-cu.ac.jp (M.M.); borage60@yahoo.co.jp (T.K.); k-ojka@daidohp.or.jp (K.O.); 2Department of Biofunctional Evaluation, Molecular Pharmacology, Gifu Pharmaceutical University, 1-25-4 Daigaku-nishi, Gifu, Gifu 501-1196, Japan; yoshino.yakkou@gmail.com (Y.Y.); shimazawa@gifu-pu.ac.jp (M.S.); hidehara@gifu-pu.ac.jp (H.H.); 3Department of Neurophysiology and Brain Science, Nagoya City University, 1 Kawasumi, Mizuho-cho, Mizuho-ku, Nagoya, Aichi 467-8601, Japan; hhida@med.nagoya-cu.ac.jp; 4Department of Neuropathology, Institute for Medical Science of Aging, Aichi Medical University, 1-1 Yazakokarimata, Nagakute, Aichi 480-1195, Japan; myoshida@aichi-med-u.ac.jp; 5Center of Excellence for Aging and Brain Repair, Department of Neurosurgery and Brain Repair, University of South Florida, Morsani College of Medicine, 12901 Bruce B. Downs Blvd, Tampa, FL 33612, USA

**Keywords:** hippocampal cholinergic neurostimulating peptide (HCNP), hippocampal cholinergic neurostimulating peptide precursor protein (HCNP-pp), hippocampal cholinergic neurostimulating peptide precursor protein knockout mouse (HCNP-pp KO mouse), cholineacetyltransferase (ChAT), local field potential (LFP), theta oscillation, Raf kinase inhibitory protein (RKIP), phosphatidylethanolamine-binding protein 1 (PEBP1)

## Abstract

The cholinergic efferent network from the medial septal nucleus to the hippocampus has an important role in learning and memory processes. This cholinergic projection can generate theta oscillations in the hippocampus to efficiently encode novel information. Hippocampal cholinergic neurostimulating peptide (HCNP) induces acetylcholine synthesis in medial septal nuclei. HCNP is processed from the N-terminal region of a 186 amino acid, 21 kD HCNP precursor protein called HCNP-pp (also known as Raf kinase inhibitory protein (RKIP) and phosphatidylethanolamine-binding protein 1 (PEBP1)). In this study, we generated HCNP-pp knockout (KO) mice and assessed their cholinergic septo-hippocampal projection, local field potentials in CA1, and behavioral phenotypes. No significant behavioral phenotype was observed in HCNP-pp KO mice. However, theta power in the CA1 of HCNP-pp KO mice was significantly reduced because of fewer cholineacetyltransferase-positive axons in the CA1 stratum oriens. These observations indicated disruption of cholinergic activity in the septo-hippocampal network. Our study demonstrates that HCNP may be a cholinergic regulator in the septo-hippocampal network.

## 1. Introduction

Theta rhythm in the hippocampal local field potential (LFP) is one of three major types of rhythm and plays a crucial role in memory processing [1]. The cholinergic septo-hippocampal system consisting of cholinergic efferents from the medial septal nucleus and diagonal band of the Broca complex (MSDB) to the stratum oriens of the CA1–CA3 area functions in hippocampal neuronal activation [2,3,4]. The cholinergic projection from the septal nucleus to the hippocampus is important as the pacemaker of theta oscillations [5], promoting more efficient encoding of novel information within episodic memory [6,7].

Hippocampal cholinergic neurostimulating peptide (HCNP), which induces acetylcholine synthesis in medial septal nuclei, was purified from the soluble fraction of the young adult rat hippocampus in 1992 [8]. HCNP is processed from the N-terminal region of a 186 amino acid, 21 kD HCNP precursor protein (HCNP-pp) [9], also known as Raf kinase inhibitory protein (RKIP) and phosphatidylethanolamine-binding protein 1 (PEBP1). HCNP facilitates the production of cholineacetyltransferase (ChAT) and possibly the synthesis of acetylcholine [9], which prevents the suppression of hippocampal glutamatergic neuronal activity induced by synthetic Aβ oligomers through the muscarinic M1 receptor [4]. In this study, we generated HCNP-pp conditional knockout (HCNP-pp KO) mice using Cre recombinase fused to a mutated ligand-binding domain of the human oestrogen receptor (ERT), which was driven by the calmodulin kinase II (CaMKII) promotor. To examine whether HCNP is involved in the hippocampal network, we investigated the integrity of the cholinergic projection, LFPs in the CA1 pyramidal layer, and memory, anxiety, and depression-related behaviors in HCNP-pp KO mice. 

## 2. Results

### 2.1. Generation of HCNP-pp KO Mice

We constructed a targeting vector in which two loxP sequences were inserted into the *HCNP-pp* gene (Accession number: AB046417) consisting of four exons. Based on our previous information, the first loxP sequence was placed on the 5′untranslated region, leaving 12 nucleotides from the start codon, in the first exon and the second loxP and neomycin-residence gene (*Neo*) were introduced into the intron that lies between exons 3 and 4. The *Neo* gene was flanked by two flippase recombinase target (FRT) sequences (Figure 1A). 

Embryonic stem cells were transfected with the linearized targeting vector and tested for recombination by southern blotting. After removing the Neo cassette by flippase and confirming the genetic sequence, correctly targeted embryonic stem cells were then injected into blastocysts to produce chimeric mice that were then crossed with wild-type C57BL/6 mice (Japan SLC, Shizuoka, Japan). Mice heterozygous for the loxP-HCNP-pp-loxP sequence (named floxed HCNP-pp: fHCNP-pp) were generated by standard procedures [10]. 

As the first step, we generated heterozygous Cre-fHCNP-pp mice (CreERT/+, fHCNP-pp/+) from homozygous fHCNP-pp mice (fHCNP-pp+/+) and heterozygous calmodulin-dependent kinase II (CaMKII) promoter-driven Cre-fused ERT transgenic mice (B6; 129S6-Tg(Camk2a-cre/ERT2)1Aibs/J, The Jackson Laboratory, ME) (CreERT/+). Next, we crossed heterozygous Cre-fHCNP-pp (CreERT/+, fHCNP-pp/+) mice with homozygous fHCNP-pp mice (fHCNP-pp+/+) to obtain homozygous Cre-fHCNP-pp mice (CreERT/+, fHCNP-pp+/+), heterozygous Cre-fHCNP-pp mice (CreERT/+, fHCNP-pp/+), and littermate control mice (CreERT/, fHCNP+/+ or /+). After daily injection of 1 mg/kg tamoxifen into the peritoneal cavity of all 3 month-old mice for 5 days, we generated heterozygous HCNP-pp KO mice and homozygous HCNP-pp KO mice with possible deletion of the genome sequence of exons 1–3 including the start codon, which was expected to encode 116 amino acids. The heterozygous and homozygous fHCNP-pp mice were viable at embryonic and perinatal stages. As confirmation of genomic deletion reaction by Cre recombinase in specific regions of the brain, the deleted allele of HCNP-pp genomic DNA were expectedly seen mainly in the frontal cortex and hippocampus, and incidentally in the cerebellum following a previous report (Appendix A) [11]. Controls included littermate control mice (CreERT/, fHCNP+/+ or /+) injected with tamoxifen, or Cre-fHCNP-pp mice (CreERT/+, fHCNP-pp+/+ or /+) injected with vehicle (corn oil).

In all experiments, we only used homozygous HCNP-pp KO (HCNP-pp KO) mice as the knockout model because the expression of HCNP-pp may be remained as a considerable amount in heterozygous HCNP-pp KO mice at the first screening analysis (Appendix A). To clarify the reduction of HCNP-pp in the hippocampus of HCNP-pp KO brains, we investigated the level of HCNP-pp in 18-month-old mice, which was 15 months after tamoxifen administration, by western blotting and immunohistochemical analysis. Western blotting revealed a significant reduction in the amount of HCNP-pp in the hippocampus of HCNP-pp KO mice compared with controls (*p* < 0.01) (Figure 1B) while a faint HCNP-pp positive band was detected. Additionally, immunohistochemical staining revealed that HCNP-pp was substantially downregulated in hippocampal pyramidal cell bodies and apical dendrites, and dentate gyrus granular cells of HCNP-pp KO mice (Figure 1C). These data showed that HCNP-pp KO mice had the expected downregulation of HCNP-pp in the hippocampus.

### 2.2. No Morphological Changes in HCNP-pp KO Mice Observed by Light Microscopy 

Next, we examined whether the reduction of HCNP-pp affected brain structures. Haematoxylin-eosin staining revealed no morphological differences between control and HCNP-pp KO brains (Figure 2A,B,G,H). 

We then carefully observed axons and myelin sheaths by light microscopy using Klüver-Barrera (Figure 2C,D,I,J) and Methenamin-Bodian (Figure 2E,F,K,L) staining. No significant differences in axons or myelin sheaths were observed in the hippocampus and medial septal nucleus between HCNP-pp KO mice and controls. These results indicated that the basic morphological structures of the hippocampus and medial septum in HCNP-pp KO mice were not different from those of controls in the analysis of light microscopy.

### 2.3. Diminished Cholinergic Projection to the CA1 Stratum Oriens in HCNP-pp KO Mice 

Neurotransmitters, including acetylcholine, maintain synaptic terminals [12]. The hippocampus receives the majority of its cholinergic inputs (up to 90%) from the medial septum via the fimbria/fornix, which enters the hippocampus through the stratum oriens [13]. Therefore, we hypothesized that the cholinergic neuronal projection into the stratum oriens may be reduced in HCNP-pp KO mice. To test this hypothesis, we performed immunohistochemistry to semi-quantify the volume of cholinergic axons in the CA1 stratum oriens using IMARIS 9.2.0 software (Bitplane, Zurich, Switzerland, see Materials and Methods). The volume of ChAT-positive axons was significantly reduced in HCNP-pp KO mice (Figure 3A,B, Control: 961.7 ± 42.5 μm^3^, *n* = 36 vs HCNP-pp KO: 637.2 ± 25.0 μm^3^, *n* = 36, *p* < 0.0001, Wilcoxson rank sum test). 

In contrast, in the secondary visual cortex (an internal control region), there was no significant difference between controls and KO mice (Figure 3A,B, Control: 786.4 ± 36.1 μm^3^, *n* = 36 vs HCNP-pp KO: 822.6 ± 34.0 μm^3^, *n* = 36, *p* = 0.417, Wilcoxson rank sum test).

### 2.4. Reduction of Theta Activity in the CA1 Pyramidal Layer of HCNP-pp KO Mice

Cholinergic MSDB neurons generate hippocampal theta rhythm via an MSDB relay [14] and their firing enhances theta oscillation in the CA1 pyramidal layer [15]. To investigate whether HCNP contributes to hippocampal functions associated with acetylcholine, we recorded LFP in the CA1 pyramidal layer (Figure 4A) of anesthetized HCNP-pp KO mice and controls, and analyzed the theta power. 

Ten controls and 12 HCNP-pp KO mice were examined. Power spectral density (PSD) was calculated from the LFP data (see Material and Methods). HCNP-pp KO mice produced significantly lower theta (3–12 Hz) power compared with controls (Figure 4B, Control: 4.11 ± 0.39 × 10^−4^ mV^2^/Hz, *n* = 69 trials vs HCNP-pp KO: 2.29 ± 0.24 × 10^−4^ mV^2^/Hz, *n* = 72 trials, *p* < 0.0001, Wilcoxson rank sum test), indicating that HCNP modulated hippocampal cholinergic activity.

### 2.5. HCNP-pp KO Mice Show No Distinct Behavioral Phenotype 

To assess spontaneous behavior, memory function, anxiety, and depression, we subjected mice to five behavioral tests: Open field, Y-maze, novel object recognition, tail suspension, and forced swimming tests.

In the open field test, the total distance travelled, and time spent in the central zone were recorded as parameters of spontaneous and anxiety-like behaviors, respectively. There was no significant difference between groups (total distance, Control: 3,564.7 ± 189.0 cm vs HCNP-pp KO: 3435.0 ± 210.1 cm, *p* = 0.65, Student’s *t*-test, Figure 5A, left, time spent in the central zone, Control: 132.8 ± 20.1 s vs HCNP-pp KO: 149.8 ± 16.9 s, *p* = 0.52, Student’s *t*-test, Figure 5A, right).

In the Y-maze test, the proportion of entries into different arms sequentially (alternation) was used as an indicator of short-term memory. The total number of entries into arms was almost equal in both groups (Control: 27.4 ± 2.4 count vs HCNP-pp KO: 27.2 ± 2.2 count, *p* = 0.95, Student’s *t*-test, Figure 5B, left). Alternation tended to be lower for HCNP-pp KO mice than controls but was not significant (Control: 68.0 ± 3.6 s vs HCNP-pp KO: 63.5 ± 1.7 s, *p* = 0.26, Student’s *t*-test, Figure 5B, right). 

In the novel object recognition test, the proportion of time spent exploring a novel object in the retention interval was used as an indicator of object recognition memory. Each exploration time tended to be shorter for HCNP-pp KO mice (Figure 5C, left), but the proportion of time exploring the novel object was almost equal in both groups (Control: 66.0 ± 2.9% vs HCNP-pp KO: 67.3 ± 4.9%, *p* = 0.82, Student’s *t*-test, Figure 5C, right).

In the tail suspension test, immobility time was used as an indicator of depression-like behavior. We measured immobile time in 2 min bins during tail suspension. A tendency to decrease immobility time in HCNP-pp KO mice was seen compared with controls, but no significance was found in the time course (*p* = 0.187, two-way ANOVA with repeated measures, Figure 5D, left) or the total immobility time (Control: 256.8 ± 9.6 s vs HCNP-pp KO: 225.6 ± 17.1 s, *p* = 0.12, Student’s *t*-test, Figure 5D, right). Furthermore, immobility time in the forced swimming test, which was similarly used as an indicator of depression-like behavior, tended to be lower for HCNP-pp KO mice than controls, but was also not significant in both the time course (time course: *p* = 0.0686, two-way ANOVA with repeated measures, Figure 5E, left) and total immobility time (Control: 247.5 ± 17.0 s vs HCNP-pp KO: 222.6 ± 21.8 s, *p* = 0.38, Student’s *t*-test, Figure 5E, right). These results indicated no significant difference in cognitive, anxiety, or depressive behaviors between controls and HCNP-pp KO mice.

## 3. Discussion

Here, we found that HCNP-pp KO mice, which had a low level of HCNP expression, exhibited a diminished cholinergic projection to CA1. Additionally, the theta activity in CA1 was reduced in HCNP-pp KO mice, which is consistent with an inhibited cholinergic projection from the MSDB to the hippocampus. However, behavioral examinations demonstrated no significant differences between controls and HCNP-pp KO mice. 

HCNP may promote acetylcholine synthesis via a quantitative increase in ChAT levels of the medial septal nucleus [8]. HCNP is cleaved from a 186 amino acid, 21 kD precursor protein by a specific enzyme of the thiol protease group [9]. In a physiological study, the amplitude of hippocampal field excitatory postsynaptic potentials produced by tetanic stimulation was markedly enhanced in HCNP-pp transgenic mice relative to controls, which was mediated through muscarinic (M1) receptor activation [3]. HCNP-pp is a multifunctional protein and is known as an ATP-binding protein, Raf kinase inhibitory protein (RKIP), and phosphatidylethanolamine-binding protein (PEBP1) [16,17]. It has an inhibitory effect on Erk signaling. Therefore, we generated conditional knockout mice using the CreERT/loxP recombination system to avoid lethality. We administrated tamoxifen at 3 months after birth because the neuronal network has been formed by this time [18]. The downregulation of HCNP-pp was confirmed by western blot analysis and immunostaining. However, a faint HCNP-pp positive band was detected by western blotting. We suggest two possible reasons for why HCNP-pp expression was not completely knocked out. The first is that HCNP-pp was downregulated by the CaMKII promotor that specifically controls expression in hippocampal excitatory neurons. We have previously reported HCNP-pp expression in inhibitory neurons, some glia cells, especially oligodendroglia, and hippocampal pyramidal neurons [19,20]. Indeed, in our model, immunostaining showed that HCNP-pp expression was maintained in some glutamic acid decarboxylase-positive interneurons (Appendix A). 

The second is that it is impossible to completely knockout a gene in all targeted cells using the CreERT/loxP system, because recombination efficiency can be adjusted by the tamoxifen dose [21]. 

HCNP is suggested to be a cholinergic modulator in the septo-hippocampal projection, which affects the maintenance of the synaptic density [3,9,12,22,23]. HCNP might also be a neurotrophic factor, because it was initially isolated from the embryonic day 14 rat hippocampus [8] and might act like nerve or fibroblast growth factor to enhance neurite outgrowth [24,25]. We, therefore, quantified the density of cholinergic axon terminals in the stratum oriens that is the main target of the cholinergic efferent projection from the MS [13]. We observed a significant decrease of cholinergic axon terminals in the stratum oriens of HCNP-pp KO mice, indicating that HCNP may facilitate maintenance of ChAT-positive terminals of septal cholinergic neurons. 

To confirm the physiological function of HCNP and/or HCNP-pp in the hippocampus, we conducted electro-physiological experiments. The cholinergic projection from the medial septal nucleus to the hippocampus may generate theta activity [1,5]. A reduced theta power was detected in the CA1 of HCNP-pp KO mice, which was consistent with the reduced cholinergic projection to the hippocampus [1,26]. 

The efficiency of the cholinergic effect from the medial septal nucleus on hippocampal theta activity might vary with the behavioral state. The theta oscillation is usually more obvious in the awake state and is associated with locomotion [27]. Indeed, acetylcholine release during activity is increased by approximately 75% compared with the inactive awake state [28]. Conversely, cholinergic activation may trigger a network effect in the septo-hippocampal system during the inactivate state but not during highly active behavior [29]. Therefore, we considered a variation in the cholinergic effect with respect to the arousal or behavioral level between individuals. The selective cholinergic effect from the medial septal nucleus on hippocampal theta activity is also stronger under anesthesia compared with the awake state [15]. About 30% of acetylcholine release is maintained under isoflurane anesthesia and the amount of acetylcholine in the hippocampus declines [30]. Based on these findings, we recorded the local field potential only under anesthesia. 

The cholinergic system in the hippocampus is well known to have a key role in memory formation and has been suggested to be involved in anxiety and depression [31]. In fact, HCNP-pp overexpression driven by the CMKII promoter produces a depressive-like phenotype [32]. A reduction of theta power in the dorsal hippocampal CA1 was expected to produce a learning, anxiety, or depression-like phenotype in HCNP-pp KO mice. However, no significant phenotype was revealed by tests of cognition or depression. Indeed, knockout of each M_1_–M_5_ muscarinic receptor in mice does not produce memory dysfunction [33]. Acquired context memory may require functions of both M_1_ and M_3_ muscarinic receptors in the dorsal hippocampus [32]. Alpha 7 nicotinic acetylcholine receptor knockout mice also display no hippocampal-dependent learning phenotype [34]. As a potential reason based on these findings, some combination of dysfunction in cholinergic receptors might be needed to display any behavioral phenotype associated with hippocampal functions and not only an acetylcholine reduction. As another reason for the negative results of the behavioral tests, ChAT activity might be sufficient under normal circumstances [35] because of other cholinergic regulators, such as nerve growth factor [36,37], TrkA and p75NTR NGF receptors [38,39,40], brain-derived neurotrophic factor [41], and neurtrophin-3/4 [42]. The deletion of TrkA, the high-affinity receptor of a major cholinergic regulator, also showed no significant behavioral abnormality even though the presence of decreased cholinergic terminals in the targeted area [43] while cognitive decline in a similar genetic TrkA deletion model was reported [44]. The regional difference of gene deletion by genetic background or Cre-line, and animal age used in study were suggested as the cause of this discrepancy [11,43]. There might be a similar problem in our model. Furthermore, behavioral tests performed in the current study cannot specifically evaluate septo-hippocampal dependent behavior. Those tests, locomotion and anxiety, working memory, cognitive memory, and depressive behavior, could assess the integrated function of several brain areas, including the hippocampus. To our knowledge, there is no specific behavioral test to evaluate septo-hippocampal cholinergic functioning while our new model may reveal specific septo-hippocampal cholinergic dysfunction. On the other hand, we recently showed that cholinergic activation via HCNP overexpression only enhances hippocampal activity under unsaturated, but not saturated, conditions [4]. Conditions that incompletely suppress hippocampal glutamatergic activation, such as under Alzheimer’s disease (AD) pathological change, might be needed to indicate the function of HCNP and/or HCNP-pp in hippocampal-related cognitive behavior [4]. 

A limitation of this study was that we could not determine whether HCNP functions via acceleration of acetylcholine release or directly by a trophic effect. Further experiments are needed to directly demonstrate the reduction of the acetylcholine concentration in the hippocampus and/or number of cholinergic neurons in the medial septal nucleus. We also could not demonstrate any phenotype-related hippocampal dysfunction by tests of cognition or depression in the current study. In the future, to confirm any phenotype of this model, we might need an investigation by an additional behavioral test or under other glutamatergic neuronal conditions, such as AD pathology.

In conclusion, we detected lower cholinergic activity in the hippocampus of HCNP-pp KO mice using immunohistochemical and electrophysiological methods, indicating that HCNP enhances cholinergic functions in the hippocampus and changes the hippocampal network in vivo. 

## 4. Materials and Methods 

### 4.1. Animals

Animal experiments were approved by the Animal Care and Use Committees of Nagoya City University Graduate School of Medical Sciences (identification code: 14-146; approval date: 23 January 2014) and conformed to guidelines for the use of laboratory animals published by the Japanese government (Law No. 105, October 1973).

The generation of HCNP-pp KO mice is described in the results section. We used 10 male mice for western blot experiments (72–74 weeks old, control (*n* = 5), homozygous HCNP-pp KO mice (*n* = 5)), 22 male mice for LFP experiments (73–90 weeks old, control (*n* = 10), homozygous HCNP-pp KO mice (*n* = 12)), and 30 male mice in behavioral experiments (56 weeks old, control (*n* = 15), homozygous HCNP-pp KO mice (*n* = 15)). The animals were housed in specific pathogen-free conditions with a 12-h light/dark cycle (lights on 08:00 to 20:00) and provided with free access to food and water.

### 4.2. Immunohistochemistry

After fixation in 4% paraformaldehyde/phosphate buffer (PB, pH 7.4), 18–22-month-old mouse brains (control: *n* = 3, HCNP-pp KO: *n* = 3) were equilibrated in a 30% sucrose solution/PB and sectioned at 20-μm thicknesses using a cryostat (Leica Microsystems, Bensheim, Germany). We generated a rabbit polyclonal anti-mouse/rat HCNP (HCNP-pp) antibody as described previously [17]. Sections were incubated overnight with the anti-HCNP-pp antibody (1:500) or a mouse monoclonal anti-GAD67 antibody (1:500, Merck-Millipore, Billerica, MA, USA) in 1% BSA/PBST at 4 °C. Bound antibodies were detected with an Alexa Fluor 488-conjugated donkey anti-rabbit IgG secondary antibody (1:500; Thermo Fisher Scientific, Waltham, MA, USA) or an Alexa Fluor 594-conjugated donkey anti-mouse IgG secondary antibody (1:500; Thermo Fisher Scientific, Waltham, MA, USA), respectively. Before mounting, brain slices were stained with DAPI (0.25 μg/mL, Sigma, St. Louis, MO, USA) for 1 min. Fluorescent signals were detected under an Axiovision fluorescence microscope (Zeiss, Oberkochen, Germany) or A1Rsi laser confocal microscope (Nikon, Tokyo, Japan). For Figure 1C, negative control was prepared to confirm a specific immunoreactive signals.

For Figure 4A, a silicon probe was coated with 2% DiI, a lipophilic fluorescent dye (D-282; Thermo Fisher Scientific, Waltham, MA, USA). Brain slices were stained with DAPI as described above.

### 4.3. Western Blot Analysis

Under deep pentobarbital anesthesia, each mouse was transcardially perfused with PBS. After their brains were removed and placed on ice, bilateral hippocampi were dissected, immediately frozen in liquid nitrogen, and stored at –80 °C until use. Frozen hippocampi from each of five HCNP-pp KO mice and five controls were homogenized in four volumes of lysis buffer containing 30 mM Tris-HCl (pH 8.5), 7 M Urea, 2 M thiourea, 4% (*w/v*) CHAPS, and a protease inhibitor cocktail (Roche Applied Science, Indianapolis, IN, USA). After incubation for 60 min on ice, homogenates were centrifuged at 15,000× *g* for 3 min at 4 °C. After the protein content was measured by the Bradford assay (Pierce, Rockford, IL, USA), the supernatants were stored at –80 °C until use. Ten micrograms of each supernatant fraction were loaded into each lane of 10% SDS-PAGE gels. After electrophoresis, samples were transferred to Hybond-P membranes (GE Healthcare, Tokyo, Japan) using 25 mM Tris, 192 mM glycine, 0.1% SDS, and 10% methanol as the transfer buffer. The membranes were then incubated with a 1:5000-diluted rabbit polyclonal anti-mouse/rat HCNP (HCNP-pp) antibody as described above or 1:50,000-diluted mouse monoclonal anti-β-actin antibody (Sigma, St. Louis, MO, USA). The membranes were then probed with HRP-conjugated anti-rabbit or anti-mouse IgG antibodies. The immunoreactive bands were visualized using an ECL Advance Western Blotting Detection kit (GE Healthcare) and recorded using an ImageQuant LAS 4000 (GE Healthcare). Western blots were quantified using Amersham Imager 600 Analysis Software (GE Healthcare).

### 4.4. Morphological Analysis

For routine histological examination, 4-μm thick, formalin-fixed, paraffin-embedded sections from hippocampal and medial-septal regions were stained with haematoxylin and eosin or by the Klüver-Barrera or Methenamin-Bodian methods. Multiple anterior-posterior tissues blocks of the hippocampus were collected from right hemispheres.

### 4.5. Quantification of Hippocampal Cholinergic Axons

We calculated the cholinergic axonal volume in 20-µm thick coronal brain sections at approximately –2.3 mm from the bregma, focusing on CA1 and the secondary visual cortex in the same section. Before ChAT immunofluorescence, each section was treated with TrueBlack Lipofuscin Autofluorescence Quencher (Biotium Inc., Hayward, CA, USA), in accordance with the manufacturer’s instructions. Briefly, TrueBlack was diluted to 1× using 70% EtOH, and sections were incubated in 1% BSA/PBS for 20 min and then in TrueBlack for 1 min, followed by rinsing in PBS. For ChAT immunofluorescence, sections were incubated overnight in 1% BSA/PBS containing a primary goat anti-ChAT polyclonal antibody (1:100; Merck-Millipore, Billerica, MA, USA). Sections were then rinsed with PBS and incubated with a secondary Alexa Fluor-594 donkey anti-goat IgG antibody (1:500; Thermo Fisher Scientific, Waltham, MA, USA) in the dark for 1 h. After rinsing in PBS, the sections were mounted on Superfrost APS-coated glass microscope slides (Matsunami Glass Industry, Osaka, Japan), allowed to dry at room temperature (RT) in the dark, and then coverslipped. Each PBS rinse step consisted of three 5-min washes, and all rinses and incubations were performed at RT on a rotating shaker. All fluorescence imaging of cholinergic fibers was performed under an A1Rsi laser confocal microscope (Nikon, Tokyo, Japan) with a 60 × 1.4 NA Plan Apochromat oil immersion lens and 4× digital zoom. Eight micron-thick Z-stacks were acquired at 0.4-μm intervals for each section within the stratum oriens layer of the CA1 field and layer II/III of the secondary visual cortex in the same slice. Images were transferred to IMARIS 9.2.0 (Bitplane, Zurich, Switzerland). To obtain estimates of the cholinergic axon volume visualized by Alexa Fluor 594 fluorescence in each region of interest, deconvoluted confocal Z-stacks were rendered in three-dimensions using IMARIS 9.2.0, and total volumes occupied by the ChAT-positive axons were calculated using an empirically optimized batch protocol that was then automatically applied to all images except for thresholds of starting and seed points that were manually optimized individually. The total volume (in μm^3^) was summed per image/per subject and then exported to an Excel spreadsheet. 

### 4.6. Surgery

All mice were anesthetized by intraperitoneal injection of ketamine (74 mg/kg) and xylazine (10 mg/kg). During surgery, mice were placed on a heating pad, and eye ointment (Tarivid ophthalmic ointment 0.3%; Santen Pharmaceutical Co., Ltd., Osaka, Japan) was used to prevent corneal drying. Local anesthesia (2% lidocaine and xylocaine jelly 2%; AstraZeneca, Osaka, Japan) was applied to the skin above the skull before making an incision to expose the skull surface. The skin was disinfected with 70% alcohol, and the skull was exposed and cleaned. Then, a head plate was firmly attached with dental cement (Fuji lute BC; GC, Tokyo, Japan, Bistite II; Tokuyama Dental, Tokyo, Japan). Mice were allowed to recover for a day before LFP recording.

For LFP experiments, dexamethasone sodium phosphate (1.32 mg/kg) was administered intraperitoneally at 1 h before surgery to prevent cerebral edema. Under 1% isoflurane anesthesia, the pericranium was scraped away and a ~2-mm diameter circular craniotomy (circle centered at 2 mm posterior and 1.3 mm lateral from the bregma) was performed over the left dorsal hippocampus. Silicon probes were inserted stereotactically, and LFP in the CA1 pyramidal layer was recorded at 4 to 10 sites per mouse. Silicon probes were coated with a 2% DiI solution (Thermo Fisher Scientific, Waltham, MA, USA) to confirm localization. 

### 4.7. Electrophysiological Recordings

Before electrophysiological recordings, mice were anesthetized with 1% isoflurane. To prevent drying of the exposed brain surface and reduce noise during recording, we applied artificial CSF (ACSF) consisting of (in mM): NaCl, 125; KCl, 3.5; NaH_2_PO_4_, 1.25; NaHCO_3_, 26; CaCl_2_, 2; MgCl_2_, 2; D-glucose, 15, to the surface of the exposed brain. Then, a 16-channel silicone probe with 177-μm^2^ recording sites (A1-16-25-177, NeuroNexus Technologies, Ann Arbor, MI, USA) was stereotactically inserted and advanced stepwise to the targeting position using a microcontroller. Pyramidal layer neural activity in the left CA1 was recorded at depths of 1100 to 1500 μm from the pia. The stereotactic coordinates were decided by referring to the atlas of Paxinos and Franklin (2008) [45]. All in vivo LFP recordings were digitally downsampled to 1 kHz using the Omniplex system (Plexon, Dallas, TX, USA) and filtered at a bandpass of 0.05 to 200 Hz. Signal pre-processing involved demeaning of the signal and 60 Hz line noise removal via a Fourier notch filter (bandwidth: <1 Hz). Proximity to the hippocampal pyramidal cell layer was judged by (1) the depth of the probe, (2) the presence of action potential discharges, and (3) the phase reversal of the LFP at theta frequencies above and below the recording site [46]. We recorded each LFP, which was presumed to be in the pyramidal layer, for 3 min. The recorded LFP data were extracted by an offline sorter (Plexon, Dallas, TX, USA), and the PSD of each LFP data set was analyzed with NeuroExplorer (Plexon). The average PSD of LFPs recorded at different electrodes at the same time was analyzed to calculate the PSD of the theta range (theta power). The theta range was defined as 3 to 12 Hz [47].

### 4.8. Behavioral Tests

All efforts were made to minimize both the suffering and number of animals used in experiments. Animals were housed at 24 ± 2 °C under a 12-h light:dark cycle (lights on from 8:00 am to 8:00 pm) and had access to food and water ad libitum. All procedures relating to animal care and treatment conformed to the Animal Care Guidelines of the Animal Experiment Committee of Gifu Pharmaceutical University (identification code: 2015-287; approval date: 25 February 2016). 

#### 4.8.1. Open Field Test

Each mouse was placed in the open field apparatus (30 cm length × 30 cm width × 30 cm height), which was made of wood, and allowed exploration of the apparatus freely for 15 min. Before commencing a new trial, the open field was cleaned with 70% ethanol and dried using paper towels. The total distance moved was recorded using a computer-operated EthoVision XT system (Noldus, Wageningen, Netherlands). The time spent in the central zone (15 cm length × 15 cm width × 15 cm height) and the number of crossings of the central zone was also recorded as markers of anxiety-like behavior. 

#### 4.8.2. Y-Maze Test

The Y-maze test was performed to assess short-term memory as described previously [48]. The Y-maze consisted of three grey plastic arms (40 cm length × 10 cm width × 12 cm height). After habituation for 1 h, each mouse was placed in the end of an arm and allowed to freely explore the maze for 8 min. During the test, behavior was recorded. The number and order of the arms entered were counted from the video file. Entering each of the three arms in turn was defined as an alternation. Alternation was calculated by the following formula:
Alternation (%) = [the number of actual alternations/the total number of arm entries − 2] × 100(1)

#### 4.8.3. Novel Object Recognition Test

The novel object recognition test is based on the innate tendency of rodents to differentially explore novel objects over familiar ones [49]. Mice were placed in the open field apparatus and habituated to the apparatus without objects for 15 min prior to the training session. At the end of each trial, the apparatus was cleaned with 70% ethanol and dried using paper towels. To exclude the effect of odor, the apparatus was then dried with a fan for 2 min. Object recognition was scored by the amount of time spent with each object (defined as the time spent with the nose of the mouse directed to the object and/or with its forelimbs touching the object). In the training session (T1: 10 min), two similar objects (left and right pyramids) were placed in symmetrical positions 5 cm away from the wall. In the retention session (T2: 10 min), two dissimilar objects were presented (a pyramid (right) and the novel object, a triangular prism (left)). During T1 and T2, the time spent exploring each object was recorded. All mice were tested following a 24-h interval between T1 and T2. Mice that did not explore the objects at all were excluded in the analysis.

#### 4.8.4. Tail Suspension Test

The tail suspension test was performed to assess depression-like behavior as described previously [31]. Each mouse was suspended by the tail with surgical tape at 50 cm above the floor, and their behavior was recorded for 8 min. Immobility time was measured automatically using a computer-operated EthoVision XT system. Mice were determined to be immobile when the mobility score of the system was less than 5%.

#### 4.8.5. Forced Swimming Test

The forced swimming test was conducted as described previously [50]. Each mouse was placed in a plastic cylinder (14-cm diameter) filled with 16 cm of water (24 ± 1 °C) for 8 min, and the immobility time was measured automatically using the computer-operated EthoVision XT system. Mice were determined to be immobile when the mobility score of the system was less than 10%.

### 4.9. Data Analysis

Data, presented as the mean ± SEM, were analyzed using the Student’s *t*-test or Wilcoxson rank sum test except for Figure 5D,E in which two-way ANOVA with repeated measures was used to analyze differences between groups.

## 5. Conclusions

HCNP-pp conditional KO displays the reduction of theta power because of fewer cholineacetyltransferase-positive axons in the CA1 stratum oriens, indicating disruption of cholinergic activity in the septo-hippocampal network. Those data suggest that HCNP may be a cholinergic regulator in the septo-hippocampal network. 

## Figures and Tables

**Figure 1 ijms-20-05367-f001:**
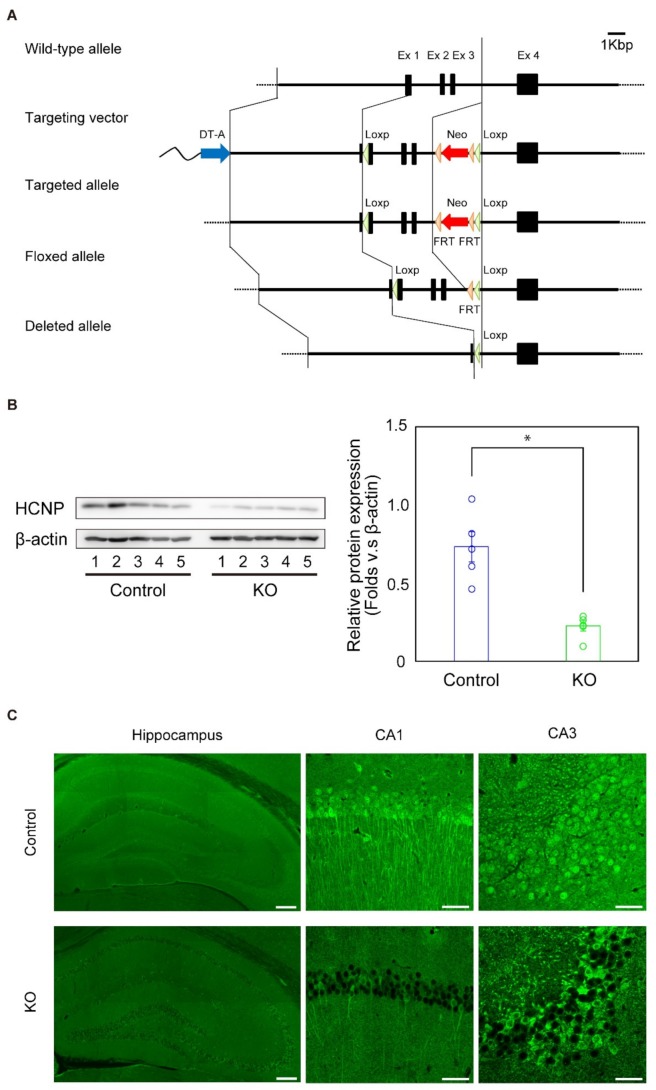
(**A**) Knockout of the *HCNP-pp* (*PEBP1*) gene by Cre recombinase. Tamoxifen was injected into floxed HCNP-pp mice, which expressed Cre-ERT, 3 months after birth. The *HCNP-pp* gene was excised between the middle portion of exon 1 and the intron between exons 3 and 4 by Cre recombinase. (**B**) Evaluation of HCNP-pp levels by western blotting (left) with quantification (right). Five controls and five HCNP-pp KO mice were examined. HCNP-pp KO mice had significantly reduced levels of HCNP-pp (Student’s *t*-test, * *p* < 0.01). Scanned images of unprocessed blots are shown in Appendix A. (**C**) Immunohistochemical staining of the hippocampus with an anti-HCNP-pp antibody. HCNP-pp expression was mainly decreased in hippocampal pyramidal cell bodies and apical dendrites and in granular cells of the dentate gyrus in HCNP-pp KO mice. Scale bar = 200 μm (left), 50 μm (middle and right).

**Figure 2 ijms-20-05367-f002:**
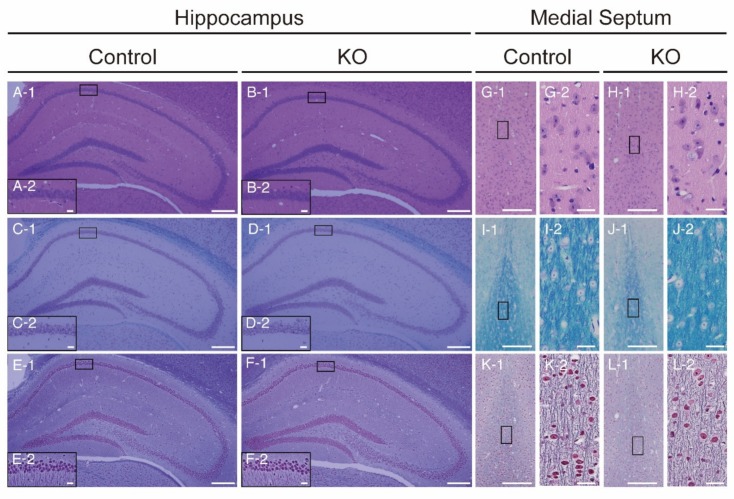
Morphological assessment of the HCNP-pp KO brain (hippocampus: **A**–**F**; medial septum: **G**–**L**). Haematoxylin-eosin staining (A-1,2, G-1,2: Control; B-1,2, H-1,2: HCNP-pp KO), Klüver–Barrera staining (C-1,2, I-1,2: Control; D-1,2, J-1,2: HCNP-pp KO) and Methenamine-Bodian staining (E-1,2, K-1,2: Control; F-1,2, L-1,2: HCNP-pp KO) revealed no morphological differences between control and HCNP-pp KO brains. Inserted boxes in all panels named 1 show areas of high magnification in each paired image, named 2 (A-2,B-2,C-2,D-2,E-2,F-2,G-2,H-2,I-2,J-2,K-2, and L-2). Scale bar = 200 μm (panels named 1) and 20 μm (panels named 2).

**Figure 3 ijms-20-05367-f003:**
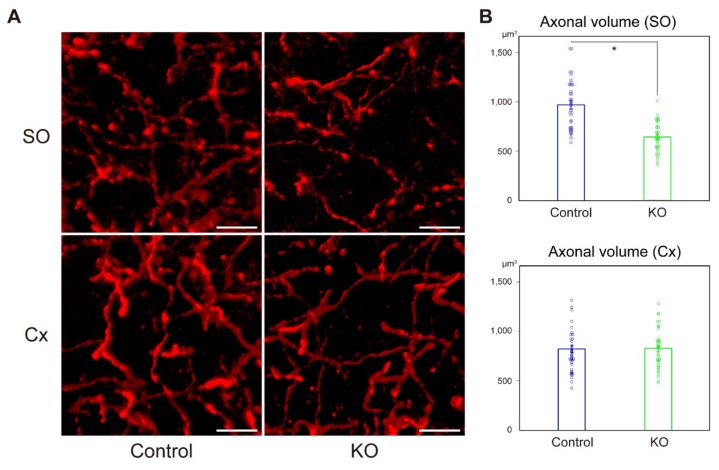
Quantification of ChAT-positive axons in stratum oriens (SO) and cortex (Cx). (**A**) Representative images of ChAT immunohistochemistry in the SO (upper) and Cx (lower) of control (left) and HCNP-pp KO (right). (**B**) Statistical analysis of the axonal volume in the stratum oriens (upper) and cortex (lower). Axonal volume in the HCNP-pp KO stratum oriens was decreased significantly (Control: 961.7 ± 42.5 μm^3^, *n* = 36 vs HCNP-pp KO: 637.2 ± 25.0 μm^3^, *n* = 36, Wilcoxson rank sum test). However, there was no significant difference in the cortex (Control; 786.4 ± 36.1 μm^3^, *n* = 36 vs HCNP-pp KO; 822.6 ±34.0 μm^3^, Wilcoxson rank sum test). Data are presented as the mean ± SEM. * *p* < 0.0001. Scale bar = 10 μm.

**Figure 4 ijms-20-05367-f004:**
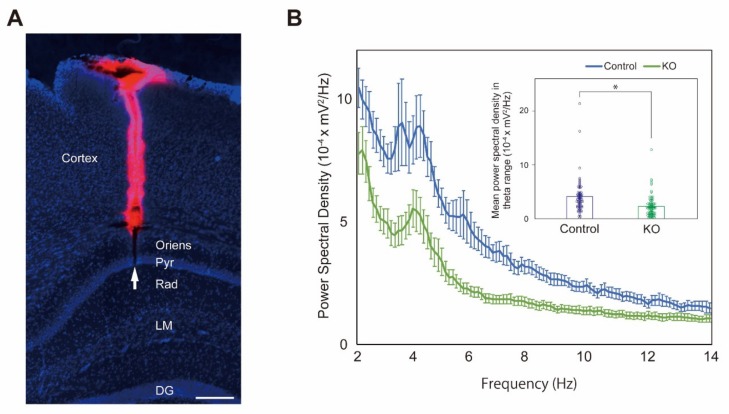
Results of electrophysiological recordings. (**A**) Probe track revealed by DiI staining in the hippocampus (scale bar: 200 μm). Pyr, stratum pyramidale; Rad, stratum radiatum, LM, stratum lacunosum-moleculare; DG, dentate gyrus; Arrow, tip of the track of the silicon probe. (**B**) Power spectral density of LFP recorded in the CA1 pyramidal layer of controls and HCNP-pp KO mice. The theta range was defined as 3–12 Hz. HCNP-pp KO mice had significantly lower theta power than controls (Control: 4.11 ± 0.39 ×10^−4^ mV^2^/Hz, *n* = 69 trials vs HCNP-pp KO: 2.29 ± 0.24 ×10^−4^ mV^2^/Hz, *n* = 72 trials, Wilcoxson rank sum test). Data are presented as the mean ± SEM. * *p* < 0.0001.

**Figure 5 ijms-20-05367-f005:**
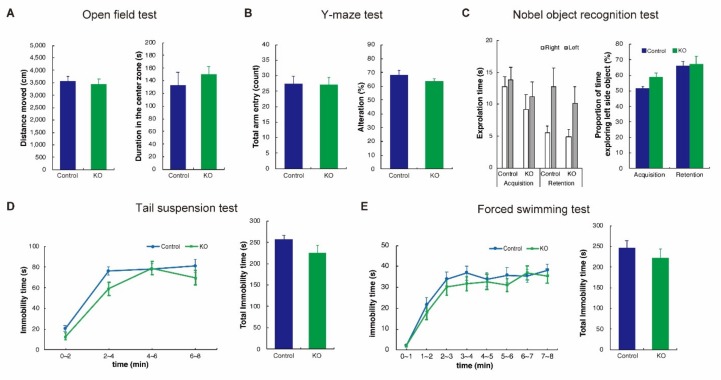
Results of behavioral examinations. (**A**) Results of open field tests. Distance moved and duration in the central zone were used as parameters of spontaneous and anxiety-like behaviors, respectively. (**B**) Results of Y-maze tests. The proportion of entries into different arms sequentially (alternation) was used as an indicator of short-term memory. (**C**) Results of novel object recognition tests. The proportion of time exploring the left side object in the retention interval was used as an indicator of object recognition memory. The retention session was performed for 24 h after the acquisition session. In the retention session, the left object used in the acquisition session was exchanged with a novel object. (**D**) Results of tail suspension tests. Immobility time was used as an indicator of depression-like behavior. (**E**) Results of forced swimming tests. Immobility time was similarly used as an indicator of depression-like behavior. In all tests, there was no significant difference between controls and HCNP-pp KO mice. Error bar represents SEM.

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
