# Peer review of "Reduced Cholinergic Activity in the Hippocampus of Hippocampal Cholinergic Neurostimulating Peptide Precursor Protein Knockout Mice"

_ijms, 2019, doi:10.3390/ijms20215367_

Round 1

Reviewer 1 Report

The article proposed for your evaluation is very interesting, but I want to  I highlight some issues:

In general, the amount of positive data is very slight, only two experiments show conclusive results and the repercussion or scientific or clinical usefulness of these data is not adequately discussed. No behavioural experiment yields positive data. The researchers suggest carrying out new behavioural tests in the future, but I consider it necessary to carry out these tests for the present article. I consider the characterization of the animal model to be incomplete. No data are shown for any technique that proves that animals are KO. A southern blot or similar could be used to verify that the animals used carry the inserted construct. I haven't been able to understand whether the control animals are being treated with tamoxifen. Tamoxifen can have effects on its own and these must be distinguishable from the KO condition. Immunofluorescence techniques are not named whether or not there is a negative control for staining. In fact, in the photographs in Figure 1, on the last panel, there is positive stainig on the animal KO that is not mentioned.

There are some minor issues:

Line 80: Stratum instead of striatum Lines 149 to 152. These definitions, in my opinion, correspond more to the section on material and methods than to the section on results. Line 163. Figure 5B is referred to when I think it refers to Figure 4B. Line 226. There's a piece of information that I can't find in the results. A feint positive band... It has never been described in the results.

It's all in my review

Best regards!

Reviewer 2 Report

The work from Madokoro et al. reports the phenotypic analysis of HCNP tamoxifen-induced KO mice, focusing on the integrity of the cholinergic septo-hippocampal system and on related behavioural readouts.

The authors show that the HCNP KO display cholinergic volume deficits in the stratum oriens of the hippocampus and reduced theta activity, as calculated by PSD. Nonetheless, the KO mice do perform as control mice in all the paradigms analysed (open field, y-maze, ORT, tail suspension and forced swimming).

The paper is written in Standard English and Results description is clear.

However, there are some theoretical and methodological considerations to take into account for an appropriate revision of the manuscript.

Major points.

1. The authors discuss the lack of behavioural phenotype in HCNP KO mice by claiming an incomplete HCNP deletion via tamoxifen treatment, and/or by NGF/TrkA-sustained compensatory mechanisms taking place in the septo-hippocampal system. However:

a) Authors did not show any southern blotting for testing homologous recombination. Protein analysis is not sufficient per se.

b) The CRE-ERT mouse strain used is not even described in the "Animals" section. Spontaneous recombinase activity has been described for the CRE-ERT2 system in vivo. This should be taken into account in data interpretation, especially when discussing an incomplete phenotype (in terms of lack of difference, as compared to the "control").

c) Opposite effects of genetic TrkA deletion on cognition have been reported, even in presence of comparable cholinergic deficits (Muller et al., 2012; Sanchez-Ortiz et al., ‎2012). Further, aging may a key factor in disturbances of forebrain-related and NGF-dependent behaviour and cognition. Current literature should be included in a more exhaustive discussion.

2. Are all the behavioural tasks examined septo-hippocampal dependent? Please, discuss.

3. The sex of the mice used for the study should be reported. 

Minor points

-group names should be consistent throughout the manuscript, figures and  figure legends (see Fig. 2).

-some typos

Round 2

Reviewer 1 Report

I believe that the authors have responded correctly to the questions raised.
I think that the proposed conditional KO mouse model may be useful for the study of a particular cholinergic pathway, but I consider it essential to further expand these studies in order to assess the usefulness of these models.
However, as the first data on this model were provided, I consider its publication to be appropriate.

Thank you for your effort

Author Response

Reviewer #1

I believe that the authors have responded correctly to the questions raised.
I think that the proposed conditional KO mouse model may be useful for the study of a particular cholinergic pathway, but I consider it essential to further expand these studies in order to assess the usefulness of these models.
However, as the first data on this model were provided, I consider its publication to be appropriate.

Response: I greatly appreciate the advice you gave us.

Reviewer 2 Report

The authors appropriately answered most of the questions, and accordingly revised the manuscript.

However, a couple of issues remains unanswered.

(Point 1a) The recombination has not been analysed in cholinergic neurons enriched basal forebrain tissues, where recombination has been already described, e.g. striatum .

(Point 1b) The issue about the control mice still exists. In fact, based on genetic background dependency of behavioural phenotypes, a CRE-ERT2 group should have been included in the study, to dissect out the effects driven by the sole presence of CRE-ERT2 allele.

The authors are encouraged to address these last points.

Author Response

Reviewer #2

I greatly appreciate your careful review. I carefully corrected the manuscript again following reviewer’s comment.

The authors appropriately answered most of the questions, and accordingly revised the manuscript. However, a couple of issues remains unanswered.

(Point 1a) The recombination has not been analysed in cholinergic neurons enriched basal forebrain tissues, where recombination has been already described, e.g. striatum .

Response: I am grateful to the reviewer's kind suggestion. In the current study, we checked the deletion of HCNP-pp genomic DNA in region of frontal cortex, hippocampus, striatum, and cerebellum of littermate control, heterozygous and homozygous HCNP-pp KO as presented in the supplemental data 2 (S2). The amount of HCNP-pp were initially checked by Western blot (S2). We previously demonstrated that soluble fraction from striatum has no effects compared to that from hippocampus, HCNP (PNAS 1984, Brain Res 1992, Prog Neurobiol 2000). We now state that future studies will examine such recombination in cholinergic neurons enriched basal forebrain tissues.

(Point 1b) The issue about the control mice still exists. In fact, based on genetic background dependency of behavioural phenotypes, a CRE-ERT2 group should have been included in the study, to dissect out the effects driven by the sole presence of CRE-ERT2 allele.

Response: I really thank you for catching up important point. Our control includes homozygous CreERT-fHCNP-pp (CreERT/+, fHCNP-pp+/+) injected with vehicle (corn oil) to dissect out the effects driven by the presence of CRE-ERT2 allele with fHCNP-pp, followed previous report (Heyward FD, J Neurosci. 36(4):1324-1335). To make the content understandable, we corrected the sentence as below in result section, page 4 line 99-101.

Controls included littermate control mice (CreERT/, fHCNP+/+ or /+) injected with tamoxifen, or Cre-fHCNP-pp mice (CreERT/+, fHCNP-pp+/+ or /+) injected with vehicle (corn oil).

Round 3

Reviewer 2 Report

The authors appropriately answered previous issues and accordingly revised the manuscript.